# Youth and Peer Mentor Led Interventions to Improve Biometric-, Nutrition, Physical Activity, and Psychosocial-Related Outcomes in Children and Adolescents: A Systematic Review

**DOI:** 10.3390/nu15122658

**Published:** 2023-06-07

**Authors:** Margaret A. Lavelle, Miriam Knopp, Carolyn W. Gunther, Laura C. Hopkins

**Affiliations:** 1Department of Communication Sciences and Disorders, College of Education and Health Sciences, Baldwin Wallace University, Berea, OH 44017, USA; mlavelle22@bw.edu; 2Department of Health Sciences, College of Education and Human Ecology, The Ohio State University, Columbus, OH 43210, USA; knopp.40@buckeyemail.osu.edu; 3College of Nursing, The Ohio State University, Columbus, OH 43210, USA; gunther.22@osu.edu; 4Department of Public Health and Prevention Science, College of Education and Health Sciences, Baldwin Wallace University, Berea, OH 44017, USA

**Keywords:** peer-led, youth-led, child, intervention

## Abstract

The utilization of youth (older) and peer (same age) mentor-led interventions to improve nutrition and physical activity has been an emerging trend in recent years. This systematic review is intended to synthesize the effectiveness of these intervention programs on participants and mentors based on biometric, nutrition, physical activity, and psychosocial outcomes of youth and peer mentor-led interventions among children and adolescents. Online databases, including PubMed, ScienceDirect, EBSCOhost and Google Scholar, were searched, and the Preferred Reporting Items for Systematic Reviews and Meta-Analyses (PRISMA) guidelines were followed. A three-step screening process was used to meet the proposed eligibility criteria, and the risk-of-bias tool for randomized trials (RoB 2) was used to assess bias for the included studies. Nineteen unique intervention programs and twenty-five total studies were deemed eligible when considering the criteria required for review. Multiple studies demonstrated positive evidence of the biometric and physical activity outcomes that were considered significant. The findings regarding the nutritional outcomes across the included studies were mixed, as some studies reported significant changes in eating habits while others did not find a significant change. Overall, the utilization of youth and peer mentor-led models in nutrition- and physical-activity-related interventions may be successful in overweight and obesity prevention efforts for those children and adolescents receiving the intervention and the youths and peers leading the interventions. More research is needed to explore the impact on the youths and peers leading the interventions and disseminating more detailed implementation strategies, e.g., training mentors would allow for advancements in the field and the replicability of approaches. Terminology: In the current youth- and peer-led nutrition and physical activity intervention literature, a varying age differential exists between the targeted sample and the peers, and varying terminology with regards to how to name or refer to the youth. In some instances, the youth mentors were individuals of the same grade as the target sample who either volunteered to serve in the peer role or were selected by their fellow students or school staff. In other cases, the youth mentors were slightly older individuals, either in high school or college, who were selected based upon their experience, leadership skills, passion for the project, or demonstration of healthy lifestyle behaviors.

## 1. Introduction

Globally, rates of childhood overweight and obesity have risen dramatically over the past three decades, and the World Health Organization’s latest reports indicate that 18% of children and adolescents worldwide are experiencing overweight or obesity. In the United States (US) [1], the overall rates of childhood obesity are high, increasing steadily over the past couple of years, despite a previous plateau [2]. According to the National Health and Nutrition Examination Survey (NHANES) data from 2017 to March 2020, obesity affected 19.7% of US youths aged 2–19 years [3]. Hispanic and Non-Hispanic Black youth are more likely to be affected by obesity than Non-Hispanic White (26.2%, 24.8%, vs. 16.6%, respectively) [3]. The detrimental effects of overweight and obesity on youth developmental, physical, mental, and social health, as well as academic progress, speaks to the gravity of this nutritional public health issue [4,5]. Additionally, dietary habits, such as fruit and vegetable consumption, are developed earlier in life and maintained throughout adulthood [6]. Therefore, effective nutrition intervention strategies promoting health behaviors must be developed and implemented, specifically targeting youth, in order to prevent obesity along with its negative consequences.

The causes and factors related to overweight, and obesity are complex, especially among children and adolescents. In addition to excess calories, various obesogenic behaviors (i.e., low physical activity, sedentary behaviors, short sleep) further promote weight gain [7]. Changing these behaviors requires an understanding of the interplay between the multilevel factors that are influencing it, which can be explained through behavior change theories [7]. A common theoretical approach utilized in community nutrition and physical activity interventions to prevent overweight and obesity is the social cognitive theory (SCT) [8,9]. According to the SCT, behavioral changes can be learned through interactions comprising personal, behavioral, and environmental factors. Peer influence (of the same age) is one example of an environmental factor influencing behavior. The concept of peer influence is not limited to the SCT, but also plays a role in social inoculation and social norms theories [10,11]. The underlying similarities in each of these theories are that peers advise each other, peers are influenced by the expectations, attitudes, and behaviors of their friend groups, and that peer influence may be stronger than an adult or professional influence [12,13]. It is well known that peer influence increases substantially during adolescence, because it is at this life stage that youth attempt to establish independence from adults [14]. Peer dynamics are such a prominent environmental influence. Therefore, interventions with child and adolescent populations should utilize peers to promote healthy behaviors [14,15]. One way to do this is through peer mentorship programs.

Peer-led interventions have been utilized among youth in areas pertaining to the use of alcohol, tobacco, illegal drugs, violence, and sexual behavior [16,17,18,19,20,21,22,23]. Data indicate that mentored youth, when compared to unmentored youth are less likely to participate in these aforementioned risky behaviors and are more likely to succeed academically [24,25,26,27,28,29]. The use of peers as an intervention strategy to improve nutrition and physical activity, and ultimately weight status, had not been employed until more recently. Two previous systematic reviews have been published: one examining peer-led nutrition education programs and the other examining peer-led physical activity education programs [30,31]. These reviews were also limited to the school setting and the training of peer mentors was analyzed in one review but overlooked in the other [30,31]. The proper training of mentors and an analysis of the training are necessary to successfully implement and disseminate an intervention.

Due to the prevalence of overweight and obesity and the risk of weight gain throughout childhood and adolescence, coupled with the increased autonomy children and adolescents experience in making diet- and physical activity-related decisions, capitalizing on peer influence to promote diet and physical activity among adolescents is crucial. As Rhodes et al. acknowledged, the research findings from peer-based interventions can be complex given the potentially confounding—characteristics of multiple individuals involved, formed relationship quality, etc. [32]. Therefore, it is the responsibility of the scientific community to successfully analyze and communicate findings to the field [32]. To the best of the authors’ knowledge, no reviews have been published synthesizing child nutrition- and physical activity-related health intervention studies that utilize a peer-led model. Thus, the objectives of this study were to:Examine the training of peers and implementation of peer roles when developing peer-led nutrition and physical activity interventions.Evaluate the extent to which peer-led nutrition and physical activity interventions had an effect on biometrics, nutrition, physical activity, and psychosocial outcomes on child and adolescent intervention participants.Evaluate to what extent peer-led nutrition and physical activity interventions had an effect on biometrics, nutrition, physical activity, and psychosocial outcomes on the peers themselves.Evaluate process outcomes of the peer-led nutrition and physical activity interventions.

It is hypothesized that peer-led nutrition and physical activity interventions will have positive impacts on child biometric, nutrition, physical activity, and psychosocial outcomes, as well as the peers’ biometric, nutrition, physical activity, and psychosocial outcomes who are delivering the intervention activities.

## 2. Materials and Methods

### 2.1. Study Design

The Preferred Reporting Items for Systematic Reviews and Meta-Analyses (PRISMA) guideline was employed during the literature review process for this study [33] An extensive search of online databases was conducted to gather articles pertaining to peer-led nutrition and physical activity interventions for children and adolescents to prevent obesity and chronic disease. PubMed, ScienceDirect, EBSCOhost, and GoogleScholar were thoroughly searched in 2018 and 2022 using key words, in varying combinations, including, but not limited to, the following: peer, peer mentor, peer mentor, youth leader, youth mentor, peer-led, mentor, mentoring, mentee, friend, nutrition, physical activity, exercise, support, social support, community, school, children, adolescent, diet, food, fruit, vegetable, prevention, health, obesity, overweight, and chronic disease. The references were also reviewed to identify additional, relevant articles. After removal of duplicates, these search procedures resulted in 667 potential studies. A schematic diagram of the PRISMA guideline is provided in Figure 1.

### 2.2. Inclusion and Exclusion Criteria

The following inclusion and exclusion criteria were used when screening the 667 articles for study eligibility. Overweight- and obesity-related health measures and behaviors, including weight, waist circumference, dietary intake and nutrition, and physical activity, were the outcomes of interest among school-aged children and adolescents (kindergarten through 12th grade). Individuals serving in “youth” or “peer” (defined below in Section 2.6) leadership roles must have been utilized in some manner during the delivery of the intervention. Studies in which the priority population had a pre-existing condition or disease, such as type 1 diabetes or cancer, were excluded. Studies in which extensive specificity of the target population limited the generalizability of the findings e.g., students attending boarding schools, were also excluded. Articles reviewing the same intervention trial were included due to the differing and contributory content of the articles. Studies presenting qualitative data were considered if the qualitative data augmented quantitative data collected for the same study.

### 2.3. Screening Protocol

A three-step screening process was conducted on the potential articles to determine article inclusion based on eligibility criteria (see Figure 1). The search and critical screening process were carried out by two authors (ML and LH). Discrepancies were discussed and a collective decision was made for inclusion or exclusion. The primary screening was conducted in which articles were eliminated due to irrelevancy based on title and a review of the abstract to identify the eligibility of the target population or outcomes of interest. This resulted in 79 potentially relevant articles. Next, a secondary screen was conducted in which articles were eliminated based on a review of the abstract and a brief scan of the entire article to match the remaining inclusion criteria, resulting in 30 potentially relevant articles. Major reasons for study exclusion during the secondary screen were limited generalizability based on the target population and the type of data presented (e.g., only qualitative data or process measures). Finally, the 30 articles were read in entirety for the tertiary screen and included if the inclusion criteria for the review were met, resulting in 25 articles addressing 19 unique studies.

### 2.4. Data Extraction

Following the tertiary screening process, the data were extracted in tabular format. Relevant information was extracted from the research articles including author(s) and year (included as the reference), study design, setting, participants, details regarding the use of youth or peer mentors, outcomes, and results. See Table 1.

### 2.5. Risk of Bias Assessment

The Cochrane risk-of-bias version 2 tool for randomized trials (RoB 2) was used to assess bias for included studies [57]. All studies were assessed with an overall bias judgement of ‘some concern’ [15,16,29,34,35,38,44,45,46,47,48,49,53,54,55,56] or ‘high risk’ [36,37,40,41,42,43,50,51,52,58]. The prevailing reason for bias was that participants or interventionists were aware of the treatment condition, which is a reality of social and behavioral intervention research. The prevailing reason that studies were assessed as ‘high risk’ was that the research designs were quasi-experimental in nature. See Table 2.

### 2.6. Definition of “Youth” and “Peer”

For the purposes of this literature review, the term ‘youth mentor’ will be used to refer to youth mentors that were older than the target sample and the term ‘peer mentor’ will be used to refer to youth mentors that were of similar age as the target sample, regardless of the terminology that was used by the investigators and authors of the interventions and studies.

## 3. Results

### 3.1. Overview of the Intervention Trials

Nineteen independent interventions were discussed in the final 25 selected articles. Table 1 contains an overview of each study included in this review, organized alphabetically by intervention trial name [See Appendix A for brief descriptions of each intervention].

The age range of the target populations as well as the settings for the interventions varied. Three of the interventions targeted elementary-school aged children, specifically children in kindergarten through sixth grade [42,43,44], while two interventions spanned the elementary-school and middle-school years [34,36]. Seven of the interventions concentrated on adolescents in middle school grades [16,38,39,40,48,49,50,52,54,55,56], while two interventions spanned both middle school and high school [35,47]. Finally, five interventions targeted solely high school-aged adolescents [15,37,41,45,46,53].

A majority of the studies were conducted in the US; however, three of the studies were conducted in Canada [37,42,43] and four were conducted in United Kingdom [48,49,51,56]. Almost all of the interventions took place either in school or after-school settings; one of the interventions took place in the homes of the participants [35]. More than half of the interventions were conducted in urban settings [29,34,35,36,37,38,39,40,41,42,43,47,50,52], several were conducted in suburban settings [15,16,38,39,48,49,51,53,54,55,56], and three were conducted in a rural setting [42,45,46]. The study designs varied across the interventions. Twelve of the studies were experimental [15,16,29,34,35,38,44,45,46,47,48,49,50,51,53,54,55,56], six were quasi-experimental [36,37,41,42,43,52], and one was a prospective observational study [40].

### 3.2. Training of Youth Mentors and Peer Mentors

Ten of the interventions utilized a youth mentor structure in which the peers were older than the target sample [29,34,36,37,40,41,42,44,51,56,58], while the other nine interventions utilized a peer mentor structure in which the peers were the same age as the target sample [15,16,38,39,43,46,48,49,50,51,52,53,54,55]. The amount of training that was provided to the youth mentors and peer mentors varied extensively, as did the amount of information and detail about the trainings. The most robust trainings and provisions of details were for the Challenge!, Fluids Used Effectively for Living (FUEL), Peer education About Weight Steadiness (PAWS), B’more Healthy Communities for Kids (BHCK), and HealthCorps interventions [29,34,35,37,41,50]. Intermediate amounts of training, spanning over several hours, and details were provided for the Healthy Hearty Kids Program (H2K), Just for Kids!, Move to Be Active (MBA), and Teens Eating for Eating for Energy and Nutrition at Schools (TEENS) interventions [16,43,44,45,46,54,55]. Finally, minimal training details were provided for the Cool Girls, Inc., Go Girls!, Girls Active Project (GAP), Healthy Buddies, MOVE Project, Peer-Led physical Activity iNterventions for Adolescent Girls (PLAN-A), Slice of Life, Students for Nutrition and eXercise (SNaX), Trying Alternative Cafeteria Options in Schools (TACOS), and the Walking in ScHools (WISH) interventions [15,36,37,38,39,40,47,48,49,53,56].

The most extensive information pertaining to the selection and training of youth mentors was provided for the Challenge! intervention. Inclusion criteria for project Challenge! mentors consisted of having a personal healthy lifestyle, expressing interest in being a role model, having experience with youth, and demonstrating commitment to the surrounding urban communities [29]. The mentee to mentor ratio was approximately 4–5:1 and a majority of the pairs were matched based on gender and race [29]. Training of mentors was approximately 40 h in length. During the training, mentors learned about the intervention, adolescent development, motivational interviewing, active listening, safety at home and in the surrounding neighborhoods, behavior management, and goal setting [29].

Similar to project Challenge!, training of youth mentors in the HealthCorps PAWS, and FUEL interventions spanned a course of weeks. Youth mentors in the HealthCorps study received a total of four weeks of training. Two weeks of the training pertained to nutrition, physical activity, and mental resilience, and the other two weeks pertained to professional development [41]. In the FUEL study, youth mentors received an “intensive” two-week training covering team building and curriculum content [37]. However, the total numbers of training hours for the HealthCorps or FUEL peer mentors were not noted [37,41].

The peer training for the H2K, Just for Kids!, PLAN-A, and TEENS, along with the interventions themselves were much shorter in length, ranging from one to two days. The peer mentors in the H2K study either volunteered or were recommended by school staff. These peer mentors underwent a 1-day, in-school training workshop that included leadership and team building training, as well as review of peer mentors’ responsibility in the H2K intervention [43]. In the Just for Kids! intervention trial, youth mentors were trained for six hours spanning a two-day period. Content of the training included learning about the intervention and responsibilities as a youth mentor, sharing viewpoints, discussing differing viewpoints as a group, motivating children, and practicing intervention delivery through role-playing [30]. In addition to pre-intervention training, youth mentors were supervised by an adult during the intervention sessions and debriefed after the sessions for reinforcement and subsequent weeks preparations [44]. Mentors in the PLAN-A study were selected by peers and participated in fifteen hours of training across 3 days [48,49]. In the TEENS study, a subset of seventh grade students were elected by their classmates to serve as peer mentors [16]. Peer mentors completed a 1-day training and their roles as peers included assisting with interactive sessions during classroom activities that were part of the intervention [16]. Similar to the HealthCorps and FUEL interventions, exact hours of training were not noted for the TEENS or H2K interventions.

Trainings of peers in the Cool Girls, Inc., Go Girls!, Healthy Buddies, MBA, MOVE Project, Slice of Life, SnaX, and TACOS interventions were extensively less compared to the aforementioned interventions. Cool Girls, Inc. participants were eligible to become youth mentors after participating in the program for one year [36]. Details of training the participants who served as mentors was not discussed [36]. In the Go Girls! intervention, youth mentors underwent training with the local Big Brothers Big Sisters agency [40]. The youth mentors In the Healthy Buddies intervention received a 45-min lesson per week from an intervention teacher then subsequently taught the same lesson to their mentees [42]. Peer mentor participants in MBA project received a four-hour training session [46]. In the MOVE Project, youth mentors attended a weekly training session immediately prior to their mentoring session [47]. The training sessions provided information about physical activity, behavior techniques, and mentoring techniques [47]. Peer mentor training in the Slice of Life intervention took place over the course of three days for a total of 16 h. The peer mentors learned about the program and engaged in role playing activities of social situations [52]. Peer mentors in the sNaX intervention attended a meeting to learn how to distribute healthy snacks and healthy messaging giveaways. No other information regarding peer mentor training was discussed [52].

Based on the articles pertaining to the studies discussing TACOS and GAP, formal training to youth and peer mentors did not seem to be provided [15,38,39,53]. In the TACOS study, mentors involved in the student groups were charged with creating promotional activities. Mentors were then classified as either low-involvement (<5 h; n = 343) or high-involvement (>5 h; n = 54) [peer mentors] based on the number of hours they dedicated to these promotional activities [15]. However, no other information was provided regarding the promotional activities or what constituted the extent of the students’ involvement [53]. In the BHCK study, extent of and format of training was not disclosed [34].

### 3.3. Child and Adolescent Participant Outcomes

#### 3.3.1. Biometric Outcomes

Various biometric outcomes were assessed for study participants in several of the interventions including BMI (prevalence of overweight and obesity), body composition, blood pressure, and heart rate. BMI and prevalence of overweight and obesity were assessed in *Challenge!*, *Just for Kids!*, *HealthCorps*, *PAWS*, *MBA* and *Healthy Buddies*. Positive impacts on weight status due to the intervention were demonstrated in project *Challenge!*, *MBA*, and the *Just for Kids!* study. There was a 5.3% decrease in overweight and obesity status among *Challenge!* participants and an 11.3% increase in overweight and obesity status among control participants (*p* = 0.02) [35]. In the *Just for Kids!* study, mean change in BMI-percentile from pre- to post-intervention was significant in the intervention group (t = −2.2; *p* = 0.03) but not for the control group (t = 0.330; *p* = 0.743) [44]. In *MBA*, reduction of BMI in intervention group was significant across follow up data collection [46]. However, in the *HealthCorps*, *PAWS*, and *Healthy Buddies* interventions, changes in BMI and prevalence of overweight or obesity did not differ significantly between the intervention and control groups [41,42].

Body composition was only assessed in three interventions—*Challenge!*, *Mentoring to be Active*, and *HealthCorps*. Findings from project *Challenge!* indicated that fat-free mass was significantly higher at 10- and 24-month follow-ups among intervention male participants compared to control male participants [35]. However, similar results were not observed for female participants. When stratified by weight status (overweight/obese vs. normal weight) significant improvements in fat percentile (*p* = 0.003), fat mass (*p* = 0.025), and fat-free mass (*p* = 0.0205) were observed in overweight/obese participants at the 24-month follow-up [35]. Compared to the teacher-led program in the *MBA* study, participants had more fat loss when program was led by peer mentors [46]. In the *HealthCorps* study, body fat percent did not differ significantly between the intervention and control groups [41].Two intervention studies, *Healthy Buddies* and *H2K*, assessed cardiovascular biometrics. Both the intervention and control groups from the *Healthy Buddies* study had increased systolic blood pressure post-intervention; however, the increase in the intervention group was significantly lower compared to the control group for both younger children and peer mentors (*p* = 0.025; *p* = 0.006) [42]. Changes in diastolic blood pressure and heart rate did not significantly differ between the intervention and control groups for the younger children [42]. In the *H2K* study, both male and female intervention participants demonstrated significantly higher VO_2_ max compared to control participants (*p* = 0.038; *p* = 0.001), meaning there was a significant gain in cardiovascular fitness among participants in peer mentoring schools [43].

#### 3.3.2. Nutrition Outcomes

##### Nutrition Outcomes: Dietary Intake

Outcomes related to dietary intake varied greatly across all studies reviewed. Only the *Challenge!*, *Slice of Life*, *BHCK*, *PAWS* and *TEENS* studies conducted a full dietary assessment. The most common dietary components that were assessed included fruits and vegetables, high or low-fat food options, and sugar-sweetened beverages (SSB) [16,29,34,35,50,51,54,55].

*Challenge!* [35] participants and *Cool Girls, Inc.* [36] participants demonstrated no significant changes related to fruit and vegetable consumption. Similarly, Cawley et al. found no significant differences in fruits and greens consumption between intervention and control groups in the *HealthCorps* studies [41]. The *Slice of Life* intervention appeared to improve overall dietary intake for females yet only improved salt intake for males [51]. *TEENS* participants exposed to classroom and school environment interventions improved their intake of fruit by a quarter serving per day (*p* = 0.052), vegetables by a quarter serving per day (*p* = 0.097), and combined fruit and vegetables by a half serving per day (*p* = 0.056) with marginal significance [16]. Youth participants in the *PAWS* study reported an increase of whole grain intake immediately and six months after intervention [50].

Methods of assessing consumption of higher or lower fat food options also varied across studies. In the *TEENS* studies, intake of higher- or lower-fat food options was assessed using food choice scores. In the *TEENS* peer mentor sub-study, participants exposed to classroom and school environment interventions significantly improved their intake of lower fat food options (*p* < 0.001), while participants exposed to only the school environment intervention improved their intake of lower fat food options with marginal significance (*p* < 0.058) [16]. In the larger *TEENS* study, intervention dose response analyses were conducted. High-dose students demonstrated significantly higher food choice scores—greater intake of lower-fat food options—compared to low-dose and control students (*p* < 0.01) [54]. Thus, greater levels of exposure resulted in significantly higher food choice scores or greater intake of lower-fat foods. In the *BHCK* study, the intervention group purchased a higher amount of healthier options (fruits, vegetables, and low-fat foods) per week compared to control (*p* = 0.01) [44]. In project *Challenge!*, snack and dessert consumption decreased significantly among intervention participants compared to control participants at 10-month (*p* = 0.001) and 24-month (*p* = 0.089) follow-ups [35]. *HealthCorps* participants, however, did not demonstrate any significant differences in fast food consumption, which is considered a higher-fat food source [41]. A final method of assessing higher or lower fat food intake in these peer-led interventions was the use of sales data. In the *TACOS* study, percentage of sales of low-fat options increased significantly in the intervention schools compared to the control schools at year 1 and year 2 (*p* = 0.096; *p* = 0.042), suggesting that participants have increased their low-fat food intake [15].

Findings related to SSB intake were fairly consistent across studies. Compared to the control participants, *HealthCorps* intervention participants significantly decreased their SSB intake by 17.5% (*p* = 0.04) [41]. Intervention participants in the *FUEL* study demonstrated a significant (*p* < 0.02) decrease in SSB consumption that sustained for three months post-intervention, whereas control participants in the *FUEL* study significantly (*p* < 0.02) increased juice consumption post intervention and continued to significantly increase juice consumption three- and twelve-months post-intervention [37]. Finally, in the *SNaX* study, SSB intake decreased for the non-peer mentor groups from 33% to 26% (*p* = 0.06) [52].

*Go Girls!* participants reported dietary behaviors with a 17-item Adolescent Food Habits Checklist [40]. Participants demonstrated a significant improvement in dietary behaviors (*p* < 0.05) from baseline to post-intervention [40].

##### Nutrition Outcomes: Nutritional Psychosocial Factors

Nutritional knowledge was assessed in three of the interventions using self-report surveys—*HealthCorps*, *Just for Kids!*, and *Slice of Life.* A 13.2% improvement in nutritional knowledge among *HealthCorps* participants compared to control participants was identified but with marginal significance (*p* = 0.09) [41]. In the *Just for Kids!* study, the intervention group demonstrated significant improvements in nutritional knowledge (*p* = 0.05) [44]. In the *Slice of Life* study, all participants appeared to improve nutritional knowledge, and females furthermore improved their food awareness [51].

Other psychosocial factors such as attitudes, perceptions, intentions, and self-efficacy were also assessed in the *Go Girls!*, *Healthy Buddies*, *Just for Kids!*, *SNaX*, and *TACOS* studies. In the *Go Girls!* study, dietary self-regulatory efficacy, intentions, and attitudes (instrumental and affective) were assessed [40]. At the post-intervention follow-up, dietary self-regulatory self-efficacy for healthy eating (*p* < 0.05) was significantly improved [40]. In the *Just for Kids!* study, theoretical concepts of intentions, attitudes, self-efficacy, and perceived support were assessed. Improvements in all these concepts for healthy eating were seen in the intervention group; however, only improved intentions and attitudes towards healthy eating were significant (*p* = 0.02; *p* = 0.05) [44]. The *SNaX* study, only assessed attitudes about the cafeteria and there were no improvements in attitudes for non-peer mentors demonstrated (*p* = 0.34) [52]. In the *TACOS* study, perceptions and behavioral intentions were assessed. Students at the *TACOS* intervention schools were significantly more likely to perceive: sale of enough low-fat foods, encouragement from adults at school to purchase low-fat foods, purchase of low-fat foods by their friends in the school cafeteria, that it was easy to identify low-fat foods in the school cafeteria, and that it was easy to purchase low-fat foods in the school cafeteria (*p* = 0.001; *p* = 0.007; *p* = 0.01; *p* = 0.03; *p* = 0.05) compared to the students at the control schools [15]. However, there were no significant differences in regards to behavioral intentions.

In the *Healthy Buddies* study, nutrition specific outcomes regarding health knowledge, behavior changes, attitudes, and body image perceptions were assessed. Regarding health knowledge, significant improvements were seen for intervention and control participants (*p* < 0.001; *p* < 0.001) and the difference between intervention and control groups was greater among intervention participants (*p* < 0.001) [42]. Similar trends were seen for health behavior and body image perceptions. Significant improvements in health behaviors were seen in the intervention (*p* < 0.001) and control (*p* < 0.001) participants; however, the difference between intervention and control groups was not significant [42]. For health attitudes assessments, significant improvements were only seen among the intervention children (*p* < 0.001), and the difference between intervention and control groups was significant (*p* = 0.043) [42].

#### 3.3.3. Physical Activity Outcomes

##### Physical Activity Outcomes: Physical Activity Behaviors

Physical activity was assessed by a variety of means including accelerometry, pedometers, fitness tests, and self-reported behaviors. The *Challenge!*, *GAP*, *PLAN-A*, and the *MOVE Project* studies utilized accelerometry technology. Among overweight or obese participants in project *Challenge!*, control participants partook in, on average, 25.5 min less physical activity daily than intervention participants (*p* = 0.018) at the 10-month follow-up time point, but no significant findings were seen at the 24-month follow-up time point [35]. In *PLAN-A*, accelerometry data showed that participants increased physical activity by 6 min compared to average compared to the control group [48,49]. The GAP study determined there was small significant difference of 2.4 min per day in mean physical activity from baseline [38,39]. No intervention effects were demonstrated with the *MOVE Project* accelerometry data [47].

The *H2K* participants reported step counts per day utilizing pedometer technology. Compared to participants at control schools, *H2K* participants at intervention schools logged significantly more steps per school day (*p* < 0.001) [43]. A significant difference was also demonstrated between gender; males logged significantly more steps per day than females (*p* < 0.05), regardless of treatment group [43].

For both the intervention and control groups in the *Healthy Buddies* study, distance covered during the 9-min run increased significantly (*p* < 0.001; *p* < 0.001); however, there was no significant difference demonstrated between treatment groups [42].

Self-reported physical activity behavior measures were utilized in the *Cool Girls, Inc.*, *Go Girls!*, and *HealthCorps*, studies. *Cool Girls, Inc.* participants reported a significant increase in physical activity behavior scores (*p* < 0.05), however, there were no additional effects among participants who were mentored and those who were not [36]. *Go Girls!* participants reported physical activity behaviors with a 2-item physical activity questionnaire and a 2-item survey for leisure-time physical activity [40]. Participants demonstrated a significant improvement in leisure-time physical activity (*p* < 0.01) from baseline to post-intervention and significant improvements in total physical activity (*p* < 0.001) and leisure-time physical activity (*p* < 0.001) at the 7-week follow-up [40]. *HealthCorps* intervention participants compared to the control participants were 45% more likely to report an increase in physical activity (*p* = 0.05) [41].

##### Physical Activity Outcomes: Physical Activity Psychosocial Factors

Physical activity psychosocial factors were assessed in the *Go Girls!*, *Just for Kids!*, *MOVE Project*, and *Slice of Life* intervention studies. In the *Go Girls!* study, physical activity self-regulatory efficacy, intentions, and instrumental and affective attitudes were assessed [40]. At the 7-week follow-up physical activity self-regulatory self-efficacy (*p* < 0.001) was significantly improved [40]. In the *Just for Kids!* study, the theoretical concepts of intentions, attitudes, self-efficacy, and perceived support for being physically active were assessed [44]. None of these concepts were significantly different for the intervention participants; however, improvements in self-efficacy for being physically active were seen among the control group (*p* = 0.05) [44]. The *MOVE Project* participants self-reported well-being, but no intervention effects were demonstrated [47]. Regarding physical activity outcomes in the *Slice of Life* study, females improved knowledge of and intentions to exercise. However, no physical activity knowledge or intentions improvements were seen among the males [51].

### 3.4. Youth and Peer Mentors Outcomes

The above outcomes discussed referred only to the participants of the studies and not the mentors themselves. Less than half of the intervention studies assessed the youth mentors or peer mentors to some extent. Two of the intervention studies, *Healthy Buddies* and *Just for Kids!*, reported on the youth mentors, and three of the intervention studies, *TACOS*, *TEENS*, and *SNaX* reported on differences between non-peer mentor participants and peer mentor participants. Data on youth mentors in the *Challenge!* study were qualitatively assessed post hoc.

#### 3.4.1. Biometric Outcomes

In the *Just for Kids!* study, youth mentor BMI-percentiles and blood pressure were assessed [45]. A marginally significant decrease in BMI was observed from baseline to post intervention (*p* < 0.06) [45]. Additionally, a medium effect size (ES = 0.56) for improved diastolic blood pressure among youth mentors was observed [45].

#### 3.4.2. Nutrition Outcomes

In the *Healthy Buddies* intervention, health knowledge, behaviors, attitudes, and perceptions of body image were assessed. Significant improvements in knowledge, health behaviors, and attitudes were seen for youth mentors (*p* < 0.001; *p* < 0.001; *p* < 0.001) and differences between youth mentor intervention and control groups were greater among the intervention group (*p* < 0.001; *p* = 0.025; *p* = 0.045) [42]. No significant improvements were seen among the youth mentor group with regards to body image perceptions [42].

In the *Just for Kids*! study, lifestyle behaviors, including a subscale for dietary behaviors, were measured [45]. A medium effect size (ES = 0.57) for improvements in dietary behaviors was demonstrated [45].

Recall, that in the *TACOS* study, no formal peer mentor training occurred. Rather, students that highly participated in student group promotion were assessed as leaders among their peers. Low-involved participants reported that *TACOS* did not change the way they chose foods and did not influence them to eat more fruits and vegetables significantly more than high-involved participants (*p* < 0.001) [53]. High-involved participants reported that their involvement in *TACOS* resulted in eating more lower-fat foods and paying more attention to what they eat significantly more than low-involved participants (*p* < 0.001) [53]. In addition to eating behaviors, significant differences were seen among all attitude, social norms, student involvement, and experience questions when comparing high-involved participants to low-involved participants (*p* < 0.001 for all questions) [53].

In the *TEENS* peer mentor sub-study, the peer mentors significantly improved their intake of fruit by a half serving per day (*p* = 0.01), combined fruit and vegetables by one serving per day (*p* = 0.012), and lower fat food options (*p* < 0.001). They also improved their intake of vegetables by a half serving per day with marginal significance (*p* = 0.059) [16]. Regarding benefits of being a peer mentor, 85% of peer mentors reported that they learned more about healthy eating and more than 64% believed that they ate healthier because of being a peer mentor [55].

In the *SNaX* intervention, attitudes about the cafeteria significantly improved among peer mentors (*p* = 0.03) [52]. When comparing the change in attitudes about the cafeteria between the peer mentors and non-peer mentors, peer mentors attitudes significantly improved compared to non-peer mentors (*p* < 0.001) [52]. Additionally, SSB intake significantly decreased for peer mentors from 33% to 21% (*p* = 0.03) [52].

#### 3.4.3. Physical Activity Outcomes

Physical activity outcomes among mentors was only measured in one study, the *Just for Kids!* Study. A significant improvement in physical activity behaviors (*p* < 0.04) was observed among youth mentors from baseline to post-intervention [45].

### 3.5. Process Evaluation

Process evaluation outcomes were only reported in four of the studies—*FUEL*, *PLAN-A*, *Slice of Life*, and *TEENS*. In the *FUEL* study, participants rated their satisfaction with the course content and delivery of the intervention on a 5-point Likert scale [54]. Overall, 77% of the participants would recommend the program to others, with significantly higher levels of satisfaction among participants in the intervention classrooms (*p* < 0.05) [54].

In the *PLAN-A* study, quantitative and qualitative measures were conducted across peer mentors, participants, teachers, parents, and intervention trainers [48,49]. Based on analysis of semi structured interviews, parents, peer-supporters, and teachers discussed an increase in the participants confidence [48,49].Training satisfaction of peer mentors was determined to be sufficient based on qualitative and quantitative measures [48,49].

Three questionnaires were used to evaluate satisfaction and perceived impact of the program in the *Slice of Life* study and all question responses were on a 5-point Likert scale [51]. Overall, females were more satisfied with the program (*p* < 0.01), believed that the program impacted their eating patterns (*p* < 0.001), enjoyed the use of university staff leading the program (*p* < 0.01) more compared to males [51]. Having peer mentors teach the program scored high and there was no significant difference in the enjoyment of peer mentors teaching the program between males and females [51]. However, females viewed the peer mentors as more able to lead group discussions (*p* = 0.01) and encourage participation (*p* = 0.01) compared to males [51].

A multicomponent process evaluation, including surveys of peer, student, and teacher perceptions of the peer mentors, implementation observation, and interviews with teachers, was conducted to assess the feasibility and acceptability of the *TEENS* peer mentor sub-study [55]. In terms of feasibility of the peer-led component of the *TEENS* sub-study, peer mentors actually led the classroom portion, which was the intervention intent, 94% of the time and were able to keep the students on task 77.5% of that time [55]. Ninety percent of the peer mentors in the TEENS sub-study stated that they enjoyed being peer mentors and 80% said they would be a peer mentor again, whereas 18% said that they would not be a peer mentor again [55]. These mentors were not probed as to why they would not want to be a peer mentor again. Among students who did not serve as peer mentors, almost 58% reported that the peer mentors were helpful [55]. Approximately 93% of teachers reported that the peer mentors were helpful or very helpful; however, one teacher reported that about 10% of the peer mentors did not take the role seriously [55]. In the teacher interviews, it was mentioned that the one-day training was too long for the kids [55]. However, no suggestions were made as to how the training could be modified for improved feasibility.

## 4. Discussion

Youth- and peer mentor-led interventions have been utilized among youth in a gamut of child development areas for over a quarter of a century. But it was not until the past decade or so that the use of peer and youth-led interventions for nutrition, physical activity, and child weight status research was implemented. Efforts to synthesize the findings from these nutrition and physical activity intervention studies has been minimal. This paper was the first to provide a review of peer- and youth-led interventions targeting children and adolescents to improve biometric-, nutrition-, physical activity, and psychosocial related outcomes for childhood overweight and obesity prevention across settings. The purpose of this study was to explore the training and implementation of youth and peers in nutrition and physical activity education interventions, evaluate the impact of the interventions on biometric-, nutrition-, physical-activity, and psychosocial related outcomes on children and adolescents receiving the interventions as well as the youth and peers leading the interventions, and to evaluate process outcomes of the interventions.

The training of the youth and peer mentors varied greatly for each intervention. Descriptions of the training processes were discussed extensively for the *Challenge!*, *Just for Kids!*, *TEENS*, and *H2K* interventions only [16,29,35,43,44,45,54,55]. If it could be determined that the intervention effects seen were due to the use of either youth or peer mentors, it would be incredibly difficult to replicate and improve generalizability of the findings. Future interventions that utilized or intend to utilize mentor model intervention strategies to improve biometric-, nutrition-, and physical activity-related outcomes should extensively detail implementation approaches, including training of mentors.

In general, both behavioral and psychosocial nutritional outcomes varied across the studies. Unfortunately, inconsistencies in findings across studies, especially regarding psychosocial factors, are common due to use of non-validated and variety of assessment tools. However, positive findings were seen for participants, youth mentors and peer mentors. The most consistent improvements were seen for increasing intake of lower-fat food options or decreasing intake of higher-fat food options, SSB intake, and nutritional knowledge.

Overall, the biometric findings were positive. Significant improvements in BMI and weight status were demonstrated in the *Challenge!*, *MBA*, and *Just for Kids*! studies, and significant improvements in body composition were seen in the *Challenge!* and *HealthCorps* studies [35,41,44,45]. The *Challenge!* and *Just for Kids!* interventions were two of the three studies with the most extensive youth mentor training, which could have contributed to the success of the intervention [29,35,44,45]. Given the short nature (12 and 8 weeks) of the interventions, these significant changes in biometrics speak to the incredible potential of these interventions, particularly if approaches that ensure programmatic sustainability is in place.

All studies that assessed physical activity outcomes demonstrated significant improvements in physical activity behaviors [35,42,43,47], and differences between intervention and control participants were only insignificant in one [42] of the studies. In the *Challenge!* study, physical activity data were derived from accelerometers that participants wore for at least nine days at a time. It is plausible that survey bias may have been playing a role, in that participants knew their physical activity was being recorded, so they consequently engaged in more physical activity than they would have otherwise [35]. In the *Healthy Buddies* intervention, step counts were calculated for the school day [42]. Thus, this measure was more of a validation of the intervention itself rather than an assessment of prolonged increases in physical activity. Future studies should explore innovative and effective ways to assess sustained physical activity in child and adolescent populations. Similar to the nutritional psychosocial outcomes, the results for physical activity psychosocial outcomes were mixed. However, improved self-efficacy for physical activity was consistent across those studies in which it was measured [40,44].

All the interventions were multi-component in nature. Examples of the components from these interventions include, but are not limited to, nutrition education, physical activity, marketing and promotion, and use of youth mentors or peer mentors. While the multi-component approach is essential to elicit behavior change [58,59,60], it makes it difficult to determine which component of the intervention resulted in the intervention effect, if one is detected. Only one study, the *TEENS* peer mentor sub-study, utilized a factorial design, to determine whether or not the innovative use of peer mentors was responsible for the intervention effect [16]. This study found that peer mentors increased their fruit and vegetable intake significantly. However, these changes were not compared to participants who received classroom and school environment intervention components or to participants who received only school environment intervention components. Additionally, several factorials were missing from this design, and effects of the peer-led intervention on participants that were mentored were not able to be determined. Therefore, future studies should utilize more rigorous factorial designs to determine whether the use of peers as mentors is an effective intervention component.

In all the interventions included in this review, methods of intervention were described as they were intended to be implemented. Whether or to what extent the interventions were implemented as planned is unknown. Two studies, *PLAN-A* and *TEENS*, provided implementation fidelity measures for the peer mentor component of the intervention. According to the authors of the article, high intervention feasibility was demonstrated [55]. Conducting process evaluations is essential to report feasibility and acceptability of the intervention, as well as implementation fidelity of the intervention. In addition to feasibility, acceptability, and fidelity, process evaluation outcomes provide insight to the sustainability of intervention strategies. Therefore, in conjunction with outcome evaluations, process evaluation measures should also be assessed in future youth-led nutrition and physical activity interventions.

### 4.1. Strengths and Limitations

Overall, positive findings were demonstrated for biometric-, nutrition-, and physical activity-related outcomes in the nineteen peer-led interventions reviewed for both participants and the peers themselves. The scoping nature of the review including the examination of the child and adolescent recipients of the interventions as well as the youth and peers that delivered the interventions, as well as the inclusion of diverse settings (e.g., school, community, etc.) is a major strength of the current study. Another major strength of the interventions reviewed in this paper was the rigor of study designs employed and the ability to interpret the findings. All the studies were either quasi-experimental, randomized controlled trials, or prospective and longitudinal in design. However, a major limitation was that the number of research studies that explored the impact on the youth and peers leading the interventions was small. Therefore, conclusions that can be drawn about the impact on the youth and peers leading the interventions themselves are limited. Additionally, a meta-analysis was not performed for the current systematic literature review. Finally, all studies included in this literature review had some or high risk for bias per the risk of bias assessment that was completed. However, prevailing reasons for these biases were realities of community-based social and behavioral interventions research, i.e., participants being aware of their treatment assignment.

### 4.2. Conclusions

Public health and nutrition professionals conducting future research engaging youth and peer mentors should utilize rigorous, factorial design to determine the impact of peers alone as an effective intervention strategy. Additionally, if youth and peer mentors are determined to be effective in improving diet and physical activity among children and adolescents, successful strategies for training and employing youth mentors and peer mentors must be understood. Finally, the impact of youth and peer mentor models on the youth and peer mentors themselves need to be more thoroughly investigated in nutrition and physical activity intervention research.

## Figures and Tables

**Figure 1 nutrients-15-02658-f001:**
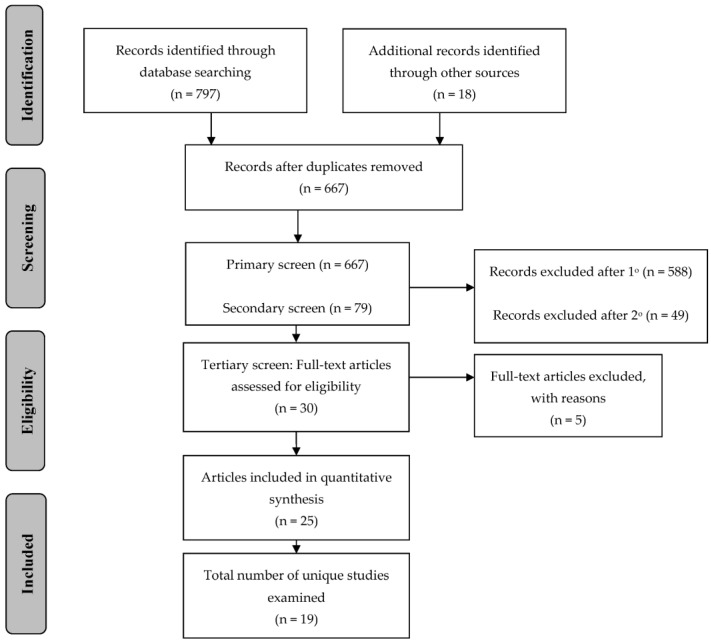
Preferred Reporting Information for Systematic Reviews and Meta Analyses (PRISMA) schematic of article selection.

**Table 1 nutrients-15-02658-t001:** Summary of Reviewed Studies Addressing Peer-Led Interventions to Improve Biometric, Nutrition, and Physical Activity Outcomes among Children and Adolescents.

Study Design	Setting	Participants	Youth Mentors or Peer Mentors	Intervention	Outcomes	Results
B’more Healthy Communities for Kids (BHCK) [34]						
Group randomized controlled trial	Thirty low-income areas in Baltimore, MD	n = 506 Aged 9–15 African American Separated into two intervention groups, 9–12 and 13–15 years old	Youth Mentors: n = 16 College Students from Baltimore City Trained by BHCK leaders	Fourteen hour-long sessions occurring every other week for 6 months	Data collection time points: baseline and post-intervention Youth food and beverage intake (fruit and vegetables) Youth food purchasing behavior	The intervention group purchased a greater variety of healthier foods per week (*p* = 0.01) No significant change in SSB and intake of fruits and vegetables
Challenge ^A^ [29,35]						
Randomized control trial	Low-income, urban neighborhoods Baltimore, MD	n = 235 Aged 11–16 years African American	Youth Mentors: n = 21 Aged 19–31 years African American	Twelve-session Home- or community-based motivational interviewing intervention by youth mentors	Data collection time points: Baseline, post-intervention (11 months), delayed follow-up (24 months) BMI z-scores and percentiles Body composition (dual wave X-ray absorptiometry; bioelectrical impedance) Physical activity (accelerometer) Diet (Youth Adolescent Food Frequency Questionnaire)	A 5.3% decrease in overweight and obesity status among participants and an 11.6% increase in overweight and obesity status among control participants (*p* = 0.006) Fat-free mass was significantly higher at 10- and 24-month follow-ups among male intervention participants Among overweight or obese participants, control participants partook in 25.5 min less daily physical activity Snack and dessert consumption decreased significantly among intervention participants (*p* = 0.001)
In-depth interviews	Low-income, urban neighborhoods Baltimore, MD	n/a	Youth Mentors: n = 17 Aged 19–31 years African American	n/a	Examination of: Mentor–mentee relationships Program impact on mentor’s behavior Future advice for peer-led programs	Establishment of mentor-mentee relationships that resulted in friendships, positive influence, and teen success Discovery of adversities in the youth’s lives Peer mentors cited personal impacts on health and life choices Peer mentors suggested a central location for sessions, the continuation of the supervisory sessions, and more programming in which teens can get actively involved
Cool Girls, Inc. [36]						
Quasi-experimental; convenience sample	Eight- schools: four elementary schools, three middle-schools, and one charter school Low-income communities Atlanta, GA	n = 175 Girls aged 9–15 years	Youth Mentors: Cool Girls, Inc. participants for <1 year	Comprehensive life skills curriculum Homework assistance Programming including weekend workshops, field trips, summer programs, etc.	Data collection time points: pre-test and post-test Healthy diet—YRBS fruit and vegetable consumption Physical activity—YRBS physical activity score	Cool Girls, Inc. participants significantly increased physical activity scores compared to control participants (*p* < 0.05) No additional effect was observed for participants matched with a peer mentor
Fluids Used Effectively for Living (FUEL) [37]						
Quasi-experimental	Three high schools Saskatchewan, Canada	n = 113 Grade 9 students	Youth Mentors: n = 7 College nutrition undergraduates (n = 5) Nutrition graduate (n = 2) Peer mentors n = 5 Same-age students selected from intervention schools	Six 45-min nutrition education lessons	Data collection time points: pre-test, post-test, 3-month and 1-year follow-up Beverage intake (beverage frequency questionnaire) Beverage intake habits Attitudes towards beverages Knowledge about healthy foods and beverages	Intervention students educated by multiple peer mentors significantly decreased their SSB intake and sustained this decrease for 3 months (*p* < 0.02) Control students significantly increased their juice intake (*p* < 0.02) Intervention participants expressed more significantly positive attitudes about the intervention compared to control participants (*p* < 0.05)
Girls Active Project [38,39]						
Cluster randomized control	Secondary state schools, Midlands, UK	n = 1752 Girls attending secondary schools aged 11–14 years	Peer Mentor: n = 56 Selected girls who were considered to have leadership qualities	Promotion of increase in physical activity in peers through an increase in physical activity culture in a school setting	Data collection time points: 0, 7 and 14 months Physical activity level Body composition Psychosocial outcomes	No difference in moderate-vigorous physical activity (MVPA) in control and intervention groups after 14 months, but a small difference (*p* = 0.05) in mean MVPA at 7 months Significant differences in self-esteem and motivation were found at 7 and 14 months A significant difference in mean acceleration and light physical activity at 7 months (*p* = 0.05) No significant difference in body composition at 7 or 14 months
Go Girls! [40]						
Prospective observational	Ontario, Canada	n = 344 Girls aged 11–14	Youth Mentors: Aged 18–25-years Women Interest in physical activity, healthy eating, and mentoring	Seven weekly, 2-h sessions Session foci: physical activity, healthy eating practices, and empowering girls Met in groups of 3–14 individuals	Data collection time points: 7-week pre-program, baseline, post-intervention, 7-week follow-up Two-item physical activity questionnaire Two-item Health Behavior in School Children survey to assess leisure-time physical activity 17-item Adolescent Food Habits Checklist Self-regulatory efficacy for physical activity Self-regulatory efficacy for healthy eating Physical activity and healthy eating intentions Physical activity and healthy eating instrumental and affective attitudes	Leisure physical activity (*p* < 0.01) and, self-regulatory efficacy for healthy eating (*p* < 0.05) were significantly improved post-intervention compared to pre-intervention Total physical activity (*p* < 0.001), leisure physical activity (*p* < 0.001), self-regulated efficacy for physical activity (*p* < 0.001), and dietary behaviors (*p* < 0.05) were significantly improved at the 7-week follow-up compared to pre-intervention
HealthCorps [41]						
Quasi-experimental	Eleven high schools In total 58% were NSLP eligible New York City, NY	n = 971 High school students	Youth Mentors: n = 6 College-graduated females White (n = 2) African American (n = 2) Hispanic (n = 2)	Physical activity and nutritional education	Data collection time points: pre-test, post-test Self-reported dietary behaviors Self-reported Physical activity Health knowledge BMI Body fat percent	SSB intake decreased, and the likelihood of participating in physical activity increased significantly in the intervention group compared to the control group (*p* < 0.05) Health knowledge improved with marginal significance in the intervention group (*p* < 0.10) Self-reported physical activity improved (*p* < 0.05)
Healthy Buddies ^A^ [42]						
Quasi-experimental	Two elementary schools British Columbia, Canada	n = 161 Kindergarten—3rd graders	Youth Mentors: n = 128 4th–7th graders	Twenty-one weekly nutrition, physical activity, and body image education sessions Two 30-min structured physical activity sessions per week	Data collection time points: beginning and the end of school year Healthy living knowledge, behavior, and attitudes Fitness (9-min run) BMI Blood pressure Heart rate	The increase in systolic blood pressure was smaller for younger children (*p* = 0.006) and youth mentors (*p* = 0.025) in the intervention group Both younger children (*p* = 0.001, *p* = 0.025, *p* = 0.035) and youth mentors (*p* = < 0.01, *p* = 0.093, *p* = 0.043) demonstrated better improvements in healthy living knowledge, behaviors, and attitudes
Heart Healthy Kids Program (H2K) [43]						
Quasi-experimental	Ten elementary schools Nova Scotia, Canada	n = 808 4th–6th graders	Peer mentors: n = 86 4th–6th grade volunteers or recommended students	Physical activity challenge Education and goal setting	Data collection time points: pre-test, post-test Physical activity (pedometer) Fitness (VO_2_ max)	The intervention participants compared to control participants demonstrated significantly higher step counts (*p* < 0.001) and VO_2_ max (males: *p* = 0.038; females: *p* = 0.001) Males, when compared to females, demonstrated significantly higher step counts (*p* < 0.05)
Just for Kids! ^A^ [44,45]						
Randomized control trial	Three elementary schools (participants) Two high schools (youth mentors) Rural Appalachian, OH	n = 72 3rd and 4th graders	Youth Mentors: High school-aged	After-school program with nutrition education and physical activity	Data collection time points: pre-test, post-test BMI-percentile Nutritional knowledge Attitudes, self-efficacy, perceived autonomy support, and intentions for eating healthy and being active	Intervention children demonstrated significant improvements in knowledge, attitudes, self-efficacy, perceived support intentions, and BMI (*p* < 0.05)
Randomized control trial	Four elementary schools (participants) Two high schools (youth mentors) Rural Appalachian, OH	n = 160 3rd and 4th graders	Youth Mentors: n = 32 10th and 11th graders	After-school program with nutrition education and physical activity Peer-delivered curriculum vs. teacher-delivered curriculum	Data collection time points: baseline and post-intervention BMI-percentile Blood pressure 5-item dietary behaviors scale 3-item physical activity behaviors scale Mediators: Knowledge, attitudes, perceived support, self-efficacy, and intentions	Peer mentored students demonstrated a significant increase in physical activity behavior (*p* < 0.04) and a marginally significant decrease in BMI-percentile (*p* < 0.06) Medium effect sizes for dietary behaviors (ES = 0.57) and diastolic blood pressure (ES = 0.56) were demonstrated for the peer mentored students
Mentoring to Be Active (MBA) ^A^ [46]						
Group randomized controlled trial	Twenty rural Appalachian high schools Southern Ohio	n = 190 Males and females aged 14–17	Peer Mentor Juniors and seniors Four-hour training session for peer mentors 10–15 mentors peer mentors per school Assigned up to 4 adolescents	Ten lesson unit over the course of 10 weeks Workbooks, homework, and weekly goal-setting involved Comparison of teacher-led vs. peer-led intervention	Data collection time points: baseline, 3- and 6-month post-intervention follow-up BMI and weight Body Fat Percentage BMI percentile for age and gender	MBA had greater difference in weight loss in obese students (*p* = 0.000) by 6 months post intervention. MBA had significant difference in weight loss of extremely obese participants (*p* < 0.05) 3 months after intervention. MBA led to a significant (*p* < 0.01) change in obese students in terms of BMI compared to the classroom 3 months after the intervention. MBA intervention in extremely obese had a significant change in BMI (*p* < 0.01) 3 months post intervention. No significant change in fat loss for extremely obese or obese participants in mentor setting 3 months after intervention. Three months post intervention, obese MBA participants experienced a significant change in body fat percentage (*p* = 0.000). Extremely obese participants did not have a significant change in body fat percentage 6 months after the intervention
MOVE Project [47]						
Cluster randomized controlled trial	Sixty schools Northern England	n = 1391 Year 7 students	Youth Mentors: 1:1 Youth Mentor: Mentee ratio Year 9 students	Youth Mentoring: 6 weekly mentoring sessions Participative learning: 6 weekly geography lessons; teacher-delivered	Data collection time points: Baseline, 6-weeks post-intervention Physical activity (accelerometer) Well-being	No significant intervention effects for physical activity or self-reported well-being (*p*-values > 0.05)
Peer-Led physical Activity iNtervention for Adolescent girls (PLAN-A) [48,49]						
Cluster randomized controlled trial	Six schools in Southwest England	n = 427 year 8 girls (aged 12–13)	Peer Mentor n = 53 Year 8 girls (12–13) Considered to be more physically active than peers Identified by peers as influential Training: 15 h spanned over 3 days Training booklet and activities	Informally discuss and provide support to peers to encourage an increase in physical activity	Data collection time points: baseline, post-intervention, and 4–5 months post-intervention Physical activity: measured with ACTi graph accelerometer Psychosocial questionnaires	No significant difference was found in the increase in the frequency of physical activity among the participants.
Process evaluation	Six schools in Southwest England	n/a	n = 53 Year-8 girls (12–13) Training: 15 h spanned over 3 days	n/a	Evaluation of the suitability of intervention, based on reports by teachers, peer-supporters, non-peer supporters, parents, and peer-supporter trainers	Peer-led intervention is likely feasible in promoting an increase in physical activity among peers.
Peer Education About Weight Steadiness (PAWS) ^A^ [50]						
Cluster randomized control	Four middle schools in east central Illinois	n = 56 Students aged 11–14	Peer mentors: n = 31 8th graders from participating schools Based on teacher references (determined to have high worth ethic) Twelve training sessions for mentors	Twelve-week course Ninety-min session separated into: 20–30 min of moderate physical activity, nutrition and cooking activities, self-reflection, goal setting for healthy eating and physical activity, discussions, and food/beverage tastings Comparison of teacher-led vs. peer-led intervention	Data collection time points: baseline, post-intervention, and 6 months post-intervention Anthropometry (BMI) Blood pressure physical activity Diet (intake of fruits, vegetables, whole grains, fats, sugar, fiber and salt) Physical activity Frequency of family meal SCT questionnaire	Self-report of whole grain intake increase from baseline to post-intervention (*p* = 0.17) and 6 months post-intervention (*p* = 0.14) Reduced calories consumed per day (*p* = 0.47) from baseline to 6 months post intervention No other significant measures reported in BMI, pressure, physical activity, and SCT questionnaire
Slice of Life [51]						
Group randomized controlled trial	One high school Suburban Minneapolis, MN	n = 270 9th graders	Peer mentors: n = 30 Student-selected peer mentors	Ten-lesson healthy eating and physical activity curriculum	Data collection time points: pre-test, post-test Dietary intake Health knowledge Health awareness Self-reported exercise Behavioral intentions	Females improved dietary intake (*p* < 0.05), health knowledge (nutrition: *p* < 0.001; physical activity: *p* < 0.05)), health awareness (*p* = 0.001), frequency of exercising (*p* < 0.05), physical activity intensity (*p* < 0.01), and physical activity intentions (*p* < 0.05) Males improved knowledge (*p* < 0.05) and salt intake (*p* < 0.05)
Students for Nutrition and eXercise (SNaX) ^A^ [52]						
Quasi-experimental	Two schools In total, 77% NSLP eligible Los Angeles, CA	n = 259 Non-peer mentors 7th graders	Peer mentors: n = 140 7th graders	Served sliced fruit instead of whole fruit Point-of-Purchase signage Handout distribution by teachers	Data collection time points: baseline, 1-month post-intervention Attitudes about the cafeteria SSB consumption	Attitudes about the cafeteria improved among peer mentors (*p* < 0.01) Peer mentors’ cafeteria attitudes improved significantly compared to non-peer mentors (*p* < 0.001) SSB intake decreased for peer mentors (*p* = 0.03)
Trying Alternative Cafeteria Options in Schools (TACOS) ^A^ [15,53]						
Group randomized control trial	Twenty high schools Suburban In total, 9% were NSLP eligible St. Paul, Minnesota	n = 1125 High school Student survey respondents	Peer mentors: Use of peer influence via student group promotion activities	Increased availability of low-fat à la carte options Peer promotion of low-fat options through school group activities	Data collection time points: baseline, Spring Year 1, Spring Year 2 Sales of low-fat à la carte items at the end of each semester Student-reported food choices Student perceptions of the food environment Students behavioral intentions	The percentage of sales of low-fat options increased significantly in the intervention schools compared to the control schools in year 1 (*p* = 0.002) and year 2 (*p* = 0.04) Perceptions of 5 components of the food environment were significantly better for students at intervention schools compared to control schools (*p*-values < 0.05)
Two-group post-test	Ten high schools Suburban In total, 9% were NSLP eligible St. Paul, Minnesota	n = 343 Less-involved students (<5 h)	Peer mentors: n = 54 Highly involved students (>5 h)	Peer promotion of low-fat options through school group activities	Perceptions of eating behaviors, attitudes, and social norms Perceptions of benefits and experiences of involvement	Significant differences were seen among all eating behaviors, attitude, social norms, student involvement, and experiences questions when comparing high-involved participants to low-involved participants (*p*-values < 0.0001)
Teens Eating for Energy and Nutrition at School (TEENS) ^A^ [16,54,55]						
Group randomized control trial; sub-study of the TEENS project	Sixteen schools In total, 20% were NSLP eligible Minneapolis and St. Paul, Minnesota	n = 3505 7th graders students	Peer mentors: n = 226 7th graders Student selected	Four intervention conditions including various components: peer mentor training, 10-session classroom lessons, school environment modifications, or none (control)	Data collection time points: beginning of school year 1, end of school year 1 Fruit and vegetable intake Usual food choices to assess selection of lower-fat options	Peer mentors demonstrated the largest and most significant increase in fruit-vegetable (*p* = 0.02) and lower-fat options (*p* = 0.02) consumption Students in the classroom and school environment condition demonstrated significant increase in low fat options consumption (*p* < 0.001) and moderately significant increases in fruit and vegetable consumption (*p* = 0.056) Students in the only school environment condition demonstrated moderately significant increases in lower fat options consumption (*p* = 0.058)
Group randomized control trial	Sixteen schools In total, 20% were NSLP eligible Minneapolis and St. Paul, Minnesota	n = 2883 7th and 8th graders	Peer mentors: n = 226 7th graders Student selected	Nutrition education Parental engagement Increasing availability of lower-fat options Developing SNACs	Data collection time points: beginning of school year 1, end of school year 1, end of school year 2 Fruit, vegetable, and energy from fat intake via 24-h dietary recall (sub-sample; n = 509) Fruit and vegetable intake and food choices via student survey	Higher levels of intervention exposure resulted in significantly higher food choice scores (*p* = 0.01)
Process evaluation of a group randomized control trial; sub-study of the TEENS project	Sixteen schools In total, 20% NSLP eligible Minneapolis and St. Paul, Minnesota	n/a	Peer Mentors: n = 226 7th graders Student selected	n/a	Teacher, peer mentor, and student perceptions of the peer mentors Feasibility of implementation	The use of peer mentors in nutrition intervention was highly feasible and accepted by the peer mentors, students, and teachers
Walking In ScHools (WISH) ^A^ [56]						
Randomized control trial	Six post-primary schools Northern Ireland	n = 187 Females aged 11–13	Youth Mentors Aged 15–17	Twelve-week period 10–15-min walks provided during the school day (8:30 a.m.–4 p.m.)	Data collection time points: baseline, end of intervention, 6-month follow-up Physical activity post intervention -Time (accelerometer) Anthropometry (weight, waist circumference, BMI) Cardiorespiratory fitness Psychosocial methods	Significant difference in light intensity of PA during school day (*p* = 0.003) of participants who participated in intervention compared to control No other significant interactions for other measured data (weight, waist circumference, BMI, (*p* > 0.005)

^A^ Indicates study in which youth mentors’ or peer mentors’ outcomes were assessed. BMI = Body Mass Index; ES = Effect Size; NSLP = National School Lunch Program; SCT = Social Cognitive Theory; YRBS = Youth Risk Behavior Survey; SSB = sugar-sweetened beverage.

**Table 2 nutrients-15-02658-t002:** Risk of Bias Assessment for Included Studies using RoB 2 [57].

Study	Randomization Process	Deviations from Intended Intervention (Intervention Assignment)	Deviations from Intended Intervention (Intervention Adherence)	Missing Outcome Data	Measurement of Outcome	Selection of Reported Results	Overall Risk of Bias Judgement
B’more Healthy Communities for Kids (BHCK) [34]	Low Risk	Low Risk	Low Risk	Some Concerns	Some Concerns	Low Risk	Some Concerns
Challenge! [29,35]	Low Risk	Some Concerns	Low Risk	Some Concerns	Low Risk	Low Risk	Some Concerns
Cool Girls, Inc. [36]	Higk Risk	Some Concerns	Some Concerns	Low Risk	Low Risk	Low Risk	High Risk
Fluids Used Effectively for Living (FUEL) [37]	High Risk	Some Concerns	Some Concerns	Low Risk	Low Risk	Low Risk	High Risk
Girls Active Project [38,39]	Low Risk	Some Concerns	Some Concerns	Low Risk	Low Risk	Low Risk	Some Concerns
Go Girls! [40]	High Risk	Some Concerns	Some Concerns	Low Risk	Some Concerns	Low Risk	High Risk
HealthCorps [41]	High Risk	Some Concerns	Some Concerns	High Risk	Low Risk	Low Risk	High Risk
Healthy Buddies [42]	High Risk	Some Concerns	Some Concerns	Low Risk	Low Risk	Low Risk	High Risk
Heart Healthy Kids Program (H2K) [43]	High Risk	Some Concerns	Some Concerns	Some Concerns	Low Risk	Low Risk	High Risk
Just for Kids! [44,45]	Low Risk	Some Concerns	Some Concerns	Low Risk	Low Risk	Low Risk	Some Concerns
Mentoring to Be Active (MBA) [46]	Low Risk	Some Concerns	Some Concerns	Low Risk	Low Risk	Low Risk	Some Concerns
MOVE Project [47]	Low Risk	Some Concerns	Some Concerns	Some Concerns	Low Risk	Low Risk	Some Concerns
Peer-Led physical Activity iNtervention for Adolescent girls (PLAN-A) [48,49]	Low Risk	Some Concerns	Some Concerns	Low Risk	Low Risk	Low Risk	Some Concerns
Peer Education About Weight Steadiness (PAWS) [50]	Low Risk	Some Concerns	Low Risk	High Risk	Low Risk	Low Risk	High Risk
Slice of Life [51]	Some Concerns	Some Concerns	High Risk	Low Risk	Low Risk	Low Risk	High Risk
Students for Nutrition and eXercise (SNaX) [52]	High Risk	Some Concerns	Some Concerns	Low Risk	Low Risk	Low Risk	High Risk
Trying Alternative Cafeteria Options in Schools (TACOS) [15,53]	Some Concerns	Some Concerns	Some Concerns	Low Risk	Low Risk	Low Risk	Some Concerns
Teens Eating for Energy and Nutrition at School (TEENS) [16,54,55]	Some Concerns	Some Concerns	Some Concerns	Low Risk	Low Risk	Low Risk	Some Concerns
WISH [56]	Low Risk	Some Concerns	Some Concerns	Low Risk	Some Concerns	Low Risk	Some Concerns

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
