# Peer review of "Youth and Peer Mentor Led Interventions to Improve Biometric-, Nutrition, Physical Activity, and Psychosocial-Related Outcomes in Children and Adolescents: A Systematic Review"

_nutrients, 2023, doi:10.3390/nu15122658_

Round 1
Reviewer 1 Report
The manuscript titled “Youth and Peer-led Interventions to Improve Biometric-, Nutrition, and Physical Activity-Related Outcomes in Children and Adolescents: A Systematic Review”.
There are several inaccuracies in the document.
The registration for the systematic review is absent. (PROSPERO)
The objectives are not concise or clear.
The discussion is unclear.
I could not find the conclusion of the manuscript.
References are used twice.
Reviewer 2 Report
The review elaborated by the authors approaches young health's current issues. The paper could be used as a reference for developing future possible educational programs in this direction. Some aspects could be considered for its improvement:
· The author's affiliation is missing.
· The journal template should be regarded as total. Some sentences have no end.
· Systematically review the English style and phrases form. There is an excessive use of the same words in a single quotation.
· The 3.1 section ”Overview of the Intervention Trials” could offer an improved overview if presented in tabular form.
The paper could be considered for proceed after major changes. It has to be revised by the author(s) and resubmitted with the suggested modification specified in the reviewer’s comments.
Reviewer 3 Report
STRUCTURE
- The manuscript is correctly structured.
TITLE AND ABSTRACT
- The Abstract is properly structured, but authors can include in each section their sub-section in bold, e.g. Background: " The utilization of youth (older) and peer (same age) led interventions aiming to improve nutrition and physical activity has been an emerging trend in recent...". In the same vein, materials and method, results and conclusions would be indicated.
- Line 14: Change this sentence “Online databases were searched to find youth and peer led programs that promoted healthy nutrition and physical activity” to “These articles were in 2022 in PubMed, ScienceDirect, EBSCOhost and Google-Scholar databases. The research used the PRISMA guidelines (preferred reporting elements for systematic reviews and meta-analyses)”.
- Results: The results must be indicated in the same way for all the observed parameters. It can indicate percentages.
- The registration number of the systematic review is not included. Review the PRISMA guidance proposal.
INTRODUCTION
- It is recommended to end the last paragraph of the introduction by stating the hypothesis under study in this work.
- Line 37: “The overall rates of childhood obesity in the US are high, increasing steadily over the past couple of years, despite a previous plateau”. Why are childhood obesity rates only shown for the US? It is recommended that global figures be added.
o World Health Organization. Obesity and overweight. Available online: https://www.who.int/es/news-room/fact-sheets/detail/obesity-and-overweight
- In general, the introduction is quite comprehensive, and shows a good overview of the current state of the subject. However, it is recommended that the authors update some references, in particular those that are more than 5 years old.
o Gelman R, Whelan J, Spiteri S, Duric D, Oakhill W, Cassar S, Love P. Adoption, implementation, and sustainability of early childhood feeding, nutrition and active play interventions in real-world settings: a systematic review. Int J Behav Nutr Phys Act. 2023 Mar 20;20(1):32. doi: 10.1186/s12966-023-01433-1. PMID: 36941649; PMCID: PMC10029282.
o Liu D, Liu D, Fan H, Tao S. Different effects of training intensity on systolic blood pressure in overweight and obese children and adolescents: A systematic review and meta-analysis. Asian J Surg. 2022 Nov 10:S1015-9584(22)01516-0. doi: 10.1016/j.asjsur.2022.10.082. Epub ahead of print. PMID: 36372706.
- Line 91: The objectives should also include nutritional psychosocial factors, which are also analysed.
MATERIAL AND METHODS
2.1. Study Design
- Does this systematic review have a registration number in the PROSPERO platform?
- First, before defining the peers, the design of this study must be specified.
- Subsequently, the following sub-section would be added: 2.2. Eligibility Criteria (line 122). It is important to separate the information to make it easier for the reader to read and find the information in the study quickly. It is recommended to specify the exclusion criteria established following the selection protocol based on the Population, Intervention, Comparison and Outcome (PICO) questions.
- It is recommended to add the search strategy included in PubMed so that the reader can find out how the articles were retrieved, if MeSH terms were used, etc (2.3. Search Strategy). Authors are requested to provide a full description of at least one full description of at least one electronic search strategy, without this it is not possible to reproduce the search.
o Moher D, Cook DJ, Eastwood S, Olkin I, Rennie D, Stroup DF. Improving the quality of reports of meta-analyses of randomised controlled trials: the QUOROM statement. Quality of Reporting of Meta-analyses. Lancet. 1999 Nov 27;354(9193):1896-900. doi: 10.1016/s0140-6736(99)04149-5. PMID: 10584742.
- What research protocol was used for data collection? (2.4. Data collection). Mentions again the protocol and adds the authors who were in charge of this part. For example: "The search and critical reading were carried out by two authors independently (Initials of the full name).
- The methodology does not include something as simple as data synthesis, i.e. how the information was collected (whether a file was prepared to record the most relevant information on the research included - authors, year, sample number, etc-) or whether the methodological quality of the studies included was carried out.
- Risk of bias in studies: Describe the methods used to assess the risk of bias in individual studies (specify whether this was done at the study or outcome level) and how this information was used in the data synthesis.
- Specify the main summary measures (e.g., risk ratio or mean difference).
RESULTS
3.1. Overview of the Intervention Trials
- Some sections end without a full stop [.]. Check this for the rest of the document.
- The type of studies finally included should be indicated. For example: Of the X studies included in this review, X were randomized control trial [reference them], X were quasi-experimental studies, etc.
3.3 Child and Adolesent Participant Outcomes
- Line 263: the p-value could be included in brackets.
3.4.1 Biometric Outcomes
- Line 444: “or improved diastolic blood pressure among youth mentors was observed.[53]”. Quote number 53 should be placed before the full stop. Applicable for the rest of the document.
3.5 Process Evaluation
- Line 493: Review the form of punctuation used in the included article [57]. It states that the scoring was from 1-3.
DISCUSSION
- Provide a cautious overall interpretation of the results taking into account the objectives, multiplicity of analyses, results of similar studies, and other relevant evidence. As the authors only discuss their results, they do not incorporate contributions from other systematic reviews that are similar.
- Line 555: In this section, references to the articles referred to by the researchers should be added.
- Lune 578: “While the multi-component approach is essential to elicit behavior change…”. The article from which this statement originates must be cited. Justifies this assertion with bibliographical contributions to emphasise its importance.
- Other limitations, such as the use of a questionnaire, lack of additional analyses (e.g. sensitivity or subgroup analyses, meta-regression) should be provided.
REFERENCES
- The references not follow the format indicated in the journal.
o https://www.mdpi.com/journal/nutrients/instructions
- The journal from which the article originates should be indicated with the journal name abbreviated and in italics. Always following the same structure, some references cannot be abbreviated and others not. For example, the abbreviation of reference number 11 should be indicated as J Am Coll Health:
o 11. Lindsey, B.J. Peer Education: A Viewpoint and Critique, J Am Coll Health 1997, 45, 187-189, doi: 10.1080/07448481.1997.9936882
- Remove square brackets from each reference.
APPENDIX
- Figure A1. Include the reference of the flow chart used to follow the methodology used to select the items. It is recommended to add this figure in the final text of the article, in the methodology section.
- Table A1. Include the meaning of BMI, SCT, ACT, MBA, ES and BHCK below the table, as it has been done for all other abbreviations.
o Urrútia, G., & Bonfill, X. (2010). PRISMA statement: a proposal to improve the publication of systematic reviews and meta-analyses. Medicina Clínica, 135(11), 507-511. https://doi.org/10.1016/j.medcli.2010.01.015
- Table A1. References 48 and 59 are the same, the access link is the same. Please check this.
- When a table is split into two sheets, the sections must be put back in the header. Applicable to the rest of the manuscript.
Round 2
Reviewer 1 Report
The manuscript titled "Youth and Peer-led Interventions to Improve Biometric Nutrition, and Physical Activity-Related Outcomes in Children and Adolescents: A Systematic Review"
Line 537 (pee-supporters)
What is this?
Reviewer 2 Report
The authors have partially considered the recommendations previously made:
- The journal template was not entirely used.
- The English style is still deficient. There is an excessive use of the same words. Please use synonyms and/or paraphrase.
The paper could be considered for proceed after minor changes. It has to be revised by the author(s) and resubmitted with the suggested modification specified in reviewer’s comments.
Reviewer 3 Report
No further comments.
